# Non-Noble Metal Catalysts in Cathodic Oxygen Reduction Reaction of Proton Exchange Membrane Fuel Cells: Recent Advances

**DOI:** 10.3390/nano12193331

**Published:** 2022-09-24

**Authors:** Zhuo Hao, Yangyang Ma, Yisong Chen, Pei Fu, Pengyu Wang

**Affiliations:** 1School of Automobile, Chang’an University, Xi’an 710064, China; 2College of Automotive Engineering, Jilin University, Changchun 130012, China

**Keywords:** transition metal, oxygen reduction reaction, proton exchange membrane fuel cells, synthesis methods, performance

## Abstract

The oxygen reduction reaction (ORR) is one of the crucial energy conversion reactions in proton exchange membrane fuel cells (PEMFCs). Low price and remarkable catalyst performance are very important for the cathode ORR of PEMFCs. Among the various explored ORR catalysts, non-noble metals (transition metal: Fe, Co, Mn, etc.) and N co-doped C (M–N–C) ORR catalysts have drawn increasing attention due to the abundance of these resources and their low price. In this paper, the recent advances of single-atom catalysts (SACs) and double-atom catalysts (DACs) in the cathode ORR of PEMFCs is reviewed systematically, with emphasis on the synthesis methods and ORR performance of the catalysts. Finally, challenges and prospects are provided for further advancing non-noble metal catalysts in PEMFCs.

## 1. Introduction

Proton exchange membrane fuel cells (PEMFCs) have been widely used in automobiles, portable power sources, fixed equipment and other scenarios due to their superior efficiency, emissions and modularity, and have attracted the attention of governments and scientific research institutes around the world [1,2,3]. However, it will be challenging for PEMFCs to quickly achieve the goal of commercial application due to their high price, insufficient durability and low power density [4,5]. Previous studies have shown that the catalyst has a great impact on the price, life and power output of PEMFCs [6]. Additionally, the kinetics of cathodic oxygen reduction reactions (ORR) are sluggish, which greatly restricts PEMFCs’ overall performance [7]. In recent decades, researchers have performed considerable research on catalysts for improving the ORR activity of PEMFC cathodes, mostly focusing on Pt-based catalysts [8,9,10,11]. However, the world’s reserves of Pt are limited and its price is expensive [12].

According to the US Department of Energy (DOE), noble metal catalysts account for almost 60% of the cost of fuel cell systems, which has greatly hindered the commercial application of fuel cells [13]. Using non-noble metals to replace Pt in the design and preparation of catalysts has become a promising measure to reduce costs. In order to effectively overcome the cost and durability challenges of fuel cell electrocatalysts, the US DOE has set a performance target for the activity and durability of non-noble metal catalysts. Specifically, the US DOE set the 2020 activity target for non-noble group metal catalysts as 0.044 A/cm^2^ at 0.9 V_iR-free_ under 1 bar H_2_-O_2_ [14], and the 2020 target for membrane electrode durability is over 5000 h [15] with no more than 30 mV of performance loss, while minimizing costs and meeting the durability target [16]. In recent years, researchers have aimed to make the performance of the designed and prepared non-noble metal catalysts come close to or exceed the DOE’s performance target. The non-noble metal catalysts have made great progress in improving PEMFCs’ cathodic ORR activity and durability, and several review papers have been published to evaluate the progress of non-noble metal catalysts [17,18,19,20,21].

In this paper, the non-noble metal catalysts were accurately identified as transition metal–heteroatoms–carbon catalysts (TM–H–C catalysts). Because the size of the nitrogen atom and the carbon atom are similar, the stability of the carbon material will not be destroyed when nitrogen atoms are doped with the carbon material. Additionally, having appropriate nitrogen atoms doped into the carbon material will improve the overall conductivity. Moreover, the nitrogen-containing group could also better disperse the metal atoms and promote the formation of active TM–H–C catalyst sites [22,23,24]. Therefore, TM–H–C catalysts with nitrogen atoms as heteroatoms are reviewed in this paper (see Figure 1).

Due to their abundant reserves, low price and strong scalability, Fe, Co and Mn are valued by researchers [25]. Recently, various transition metal–nitrogen–carbon catalysts (TM–N–C catalysts, TM: Fe, Co, Mn, etc.) have been studied and prepared, and they have shown promising electrocatalytic activity and durability [26,27]. The main reason for the excellent performance of TM–N–C catalysts is the synergistic effect between transition metal atoms, nitrogen, and carbon materials [28]. Furthermore, with the help of spectroscopy technique and density functional theory (DFT), it was found that the active sites of atomic metal coordinated nitrogen sites (such as, Fe-N_X_, Co-N_X_ and Mn-N_X_.) was the main reason leading to the activity of TM–N–C catalysts [29,30,31]. However, the structure of TM–N–C active site is complex and may be dynamically changed during ORR process, so it is a challenge to clearly describe the reasons for the improved ORR performance [19,32].

In recent years, many non-noble metal catalysts for the cathode ORR of PEMFCs have been developed, which provide references for this paper. A total of 160 related studies were referenced in this review, of which, 78.75% were published in the last five years. The impact factors of the published studies in the last five years were also classified, as shown in Figure 2.

In order to advance the understanding and development of new high-performance non-noble metal catalysts, the research progress of non-noble metals and N co-doped carbon catalysts is extensively reviewed in this paper. However, many recent reviews have also been published [33,34,35]. In light of this, we not only subdivide non-noble catalysts into single-atom catalysts and double-atom catalysts, but also further focus on the preparation methods and performances of catalysts with Fe, Co and Mn as non-noble metal atoms. The challenges and prospects of non-noble metal catalysts used in the ORR of PEMFCs are discussed and predicted. Specifically, the purpose and main contributions of this paper include: (i) A comprehensive summary of the synthesis progress of non-noble metal catalysts (especially single-atom catalysts and double-atom catalysts) over the past five years. (ii) A presentation of the important highlights and challenges regarding the design and synthesis of non-noble metals. This review can provide better insight into current progress and future directions, and provide some reference value for related studies on the design and synthesis of non-noble metal catalysts.

## 2. Transition Metal-Nitrogen-Carbon Catalysts

Transition Metal-Nitrogen-Carbon catalysts (TM–N–C catalysts) are considered to be the most promising catalysts for cathode ORR of PEMFCs [36], and researchers have also carried out detailed and considerable research on TM–N–C catalysts. In 1964, Jasinski first reported the high-efficiency ORR catalytic action of cobalt phthalocyanine (CoPe) at room temperature [37]. However, the metal macrocyclic compounds proposed by Jasinski have the shortcomings of insufficient stability and poor electrical conductivity. Subsequent researchers reported that the overall performance of metal macrocyclic compounds can be improved by heat treatment [38,39]. In 1989, researchers successfully prepared active ORR catalyst using polymer, Co salt or Fe salt, carbon black support and other materials [40], which pioneered the preparation of ORR catalyst with low-cost materials. From then to the early 20th century, researchers proposed the use of several different transition metals, such as Fe, Co and Ni [41,42,43], as well as non-macrocyclic nitrogen source materials, such as Pyridinic type nitrogen, Cyanamide and nitrogen containing salt [44,45,46]. The TM–N–C catalysts prepared in subsequent reports had been comparable to the Pt-C catalyst. This section mainly focus on the synthesis methods and performance of single metal atom catalysts and double metal atom catalysts.

The reduction of the size of metal particles is conducive to improving the reactivity of supported metal catalysts [47]. With the development of nanotechnology, the size of metal particles could be reduced to nanoscale or sub-nanoscale [48], and some reports indicated that sub-nanoscale supported metal catalysts can exhibit better catalytic activity [49,50]. The active sites exposure rate and catalytic activity of TM–N–C catalysts can be effectively improved by further reducing the non-noble metal nanoparticles to atomic scale [51,52]. Compared with nanoscale transition metal particle catalysts, atomic scale transition metal catalysts have many advantages: (i) with unique electronic structure and definite active site, the atomic scale catalysts can exhibit excellent catalytic performance [53,54]; (ii) the atomic scale catalysts can facilitate the activation of reactants by lowering energy barrier for a high selectivity [55,56]; (iii) from the perspective of atomic scale, the structure-performance relationship of catalysts can be clearly established and understanded, and with the help of DFT theory and experiments, the position of active sites can be clearly identified, which can provide reference for the improved design of high-performance atomic level transition metal catalysts [57,58].

### 2.1. Single-Atom Catalysts (SACs)

Single-atom catalysts (SACs) can maximize the utilization rate of transition metal atoms, theoretically reaching 100% of the atom utilization rate [59]. Moreover, the spatial structure of SACs is very uniform, with an unsaturated coordination environment and clear single atom sites, which can completely expose the active sites attached to the support surface [60]. At the same time, the unique electronic structure of transition metal active center atoms effectively improves catalytic activity and selectivity, as well as improve the stability of the catalysts [61,62]. These advantages provide the premise for the wide research and application of SACs. For the new catalysts of atomic scale, researchers showed great interest in designing and preparing SACs using Fe, Co, Mn and other non-noble metal atoms.

#### 2.1.1. Fe-SACs

The most commonly used method to synthesize Fe-N-C catalysts is to mix and pyrolyze Fe precursor, N source and C matrix [63]. However, this synthesis method is complicated, and it is difficult to form a strong interaction between single atom Fe and the support, and the prepared Fe-SACs are prone to the Fenton reaction, resulting in dissolution [64,65]. Therefore, researchers tried new methods to synthesize Fe-SACs. Zheng et al. used a nitrogen rich bridging ligand (tetrapyridophenazine, tpphz) as carbon and nitrogen sources, and prepared Fe-tpphz from Fe ions and tpphz molecules under solvothermal treatment with Fe (II) [66]. Then, Fe-tpphz was pyrolyzed and etched to obtain Fe-N/C catalyst with high stability and good activity (see Figure 3a). The test and measurement results showed that the prepared Fe-N/C catalyst has excellent ORR activity and stability under acidic and alkaline conditions. Li et al. reported a method for preparing Fe-N-C catalyst by using 2-methylimidazole (2-MIM), ZnO, and ferrous oxalate (FeC_2_O_4_·2H_2_O, FeOx) powder mixture [67], which is simple, environmental friendly and low price. Concretely, Fe (II)-doped zeolitic imidazolate frameworks (ZIF-8) were first prepared, which were denoted as Fe2-Z8, and Fe2-Z8 crystals were then carbonized in Ar at 1000 °C to obtain the final Fe2-Z8-C electrocatalyst without any further treatment. The immediately available nitrogen atoms could then be firmly combined with neighboring carbon atoms to form Fe-N-C catalyst (see Figure 3b).

Iron compounds can also be used as iron precursors for the preparation of Fe-N-C catalysts, such as iron salts (FeCl_3_) [68,69] and iron oxides (Fe_2_O_3_) [70,71]. Xiao et al. reported a method of homogeneously introducing commercial Fe_2_O_3_ as a solid-state Fe source into ZIF-8 to synthesize Fe-N-C catalysts [72]. The Fe-N-C derived from the solid Fe_2_O_3_ precursor showed a porous framework without obvious particle formation, and Fe, N and C were homogeneously distributed in the Fe-N-C catalysts (see Figure 3c). The preparation method reported by Xiao et al. is facile and practicable. The half-wave potential of F-N-C catalysts prepared in acidic and alkaline electrolytes achieved 0.82 V and 0.90 V (versus reversible hydrogen electrode, vs. RHE) respectively, showing excellent ORR activity.

**Figure 3 nanomaterials-12-03331-f003:**
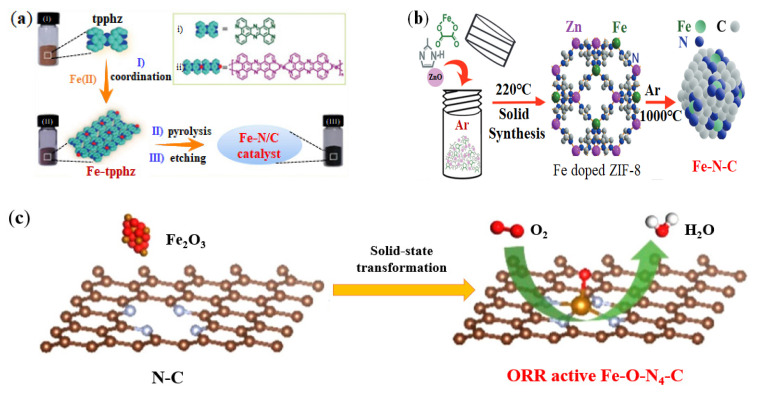
Synthesis methods of Fe-SACs: (**a**) Synthesis from Fe-tpphz complex. Adapted with permission from Ref. [66]. Copyright 2018 Royal Society of Chemistry. (**b**) Synthesis from Fe doped ZIF-8. Adapted with permission from Ref. [67]. Copyright 2018 John Wiley and Sons; (**c**) Synthesis from Fe_2_O_3_ and ZIF-8. Adapted with permission from Ref. [72]. Copyright 2021 American Chemical Society.

#### 2.1.2. Co-SACs

Compared with Fe-SACs, Co-SACs are hardly affected by Fenton reaction and have better stability in the cathode ORR of PEMFC [73,74]. Recently, the power density of fuel cells equipped with Co-SACS catalysts has also been improved, which has greatly attracted the research interest of researchers. Yin et al. reported a method for preparing stable Co single atoms (SAs) on nitrogen-doped porous carbon [75]. Concretely, the method was based on the pyrolysis process of the pre-designed bimetallic Zn/Co metal organic framework (MOF), Co was reduced by carbonization of the organic linker, and Zn   was selectively evaporated at a high temperature higher than 800 °C to synthesize Co single atoms/nitrogen doped porous carbon (Co SAs/N-C) catalysts (see Figure 4a). It is worth pointing out that MOF has been emerging as the selected precursor to synthesize SACs [76,77,78]. Especially, ZIF is a subgroup of MOF, which is also considered to be the SACs precursors [67,72,79,80]. Sun et al. reported a facile and practicable “sacrificed-template” method for preparing the cobalt single-atom electrocatalysts with urchin-like nano-tube hierarchical structures (UNT Co SAs/N-C) [81]. The three-step synthesis strategy was shown in Figure 4b: the preparation of Cobalt Carbonate Hydroxide with Urchin-like NanoRods (UNR CCH); the preparation of Urchin-like ZIF-67 (UNT ZIF-67); the preparation of UNT Co SAs/N-C catalysts.

Wan et al. reported that the ORR catalysts synthesized based on ZIF had the problems of large particle size and low mesoporous ratio, leading to poor electron conductivity and affecting the catalytic performance of ORR [82]. Therefore, increasing mesoporous rate and conductivity is an effective strategy to improve ORR catalyst [83]. Wang et al. synthesized a CoNC@KJ600 catalyst with high pore structure and high electronic conductivity based on ZIF, and used the same procedure to synthesize CoNC catalyst to compare and verify the performance of CoNC@KJ600 [84]. The synthesis procedure was shown in Figure 5a. During the synthesis process, the porous structure of KJ600 carbon black was retained, and the Co element was highly dispersed in CoNC@KJ600 catalyst. However, there were lots of Co nanoparticles in CoNC catalyst. The presence of Co nanoparticles would block the mass transfer gap and reduce the activity of ORR catalyst [85]. As shown in Figure 5b,c, the catalytic current density of CoNC@KJ600 catalyst was slightly higher than that of CoNC catalyst (1.58 vs. 1.28 A g^−1^ @ 0.8V), and CoNC@KJ600 catalyst was more durable than CoNC catalyst after 20 h test. Considering the high pore structure and high electronic conductivity of CoNC@KJ600 catalyst, CoNC@KJ600 catalyst could be applied to PEMFC. The peak power density of PEMFC with CoNC@KJ600 catalyst as cathode was 0.92 W/cm^2^, which was higher than that reported by Cheng et al. [86] and Im et al. [87] for PEMFC with Co-N-C catalyst as cathode.

Cheng et al. reported a type Co-N-CNFs catalyst that single Co and N atoms co-doped carbon nanofibers (CNFs) [88]. The test results showed that Co-N-CNFs have high durability and ideal ORR activity in both acidic and alkaline electrolytes. Meanwhile, from the structure-activity-durability relationship of Co-N-CNFs, single atom Co was more suitable to be an effective active component for the development of TM–N–C catalyst than single atom Fe. After further study, Cheng et al. reported a novel type Co@SACo-N-C catalyst that Co nanoparticles embedded in single Co and N atoms co-doped CNFs [89], and the preparation diagram of Co@SACo-N-C catalysts was shown in Figure 6a. Linear sweep voltammetry (LSV) is often used to evaluate the ORR catalytic performance of catalysts [90,91]. As shown in Figure 6b, the onset potential (E_onset_) of Co@SACo-N-C-10 catalyst was 0.92 V and the half-wave potential (E_1/2_) was 0.778 V (in 0.1 M HClO_4_ solution), which was only 0.62 mV different from the commercial Pt/C catalyst. And Co@SACo-N-C-10 catalyst’s E_1/2_ displayed only 9 mV decay after a 10,000 accelerated degradation test (ADT) cycling (see Figure 6c), which showed the excellent durability in acidic electrolytes.

#### 2.1.3. Mn-SACs

It has been reported that Mn-N-C catalysts exhibit helpful catalytic activity and is more suitable than Fe-N-C catalysts to be platinum group metals-free (PGM-free) ORR catalysts for PEMFC cathode [92,93,94]. Unlike Fe and Co atoms, Mn cannot easily exchange Zn and form a mixture with N in the precursor of ZIF-8. At the same time, during high temperature carbonization, Mn is easy to form aggregates due to its various valence states of 0~+7, which makes it difficult to synthesize Mn-N-C catalysts [95]. Li et al. reported a method of synthesizing Mn-NC catalyst using ZIF-8 precursor [96], and the catalyst with active site of MnN_4_ was obtained by two-step synthesis strategy (see Figure 7a). In the first step of synthesis, Mn ions were combined with Zn ions to synthesize MN-doped ZIF-8 precursor, and then carbonized and acid leached to obtain the best nitrogen doped and microporous carbon body. In the second step of synthesis, additional manganese and nitrogen sources were adsorbed to the carbon subject, followed by thermal activation to obtain a more active M-N-C catalyst. Liu et al. developed a method for synthesizing Mn-N-C catalyst by hydrogel polymer [97]. As shown in Figure 7b, polyaniline (PANI) was used as the carbon/nitrogen sources, and Mn^2+^ source was added in the polymerization process and evenly dispersed into the precursor of PANI hydrogel. The high temperature carbonization process was used to transform PANI-Mn hydrogel into Mn and N co-doped carbon, namely Mn-N-C catalyst. Then, followed by a second pyrolysis process to remove inactive substances and recover carbon oxide to improve catalytic activity. And in Figure 7c,d, the PANI hydrogel-derived Mn-N-C catalyst exhibited ORR activity that was similar to the Fe-N-C catalyst, and also showed excellent ORR durability.

Chen et al. reported an effective strategy for the synthesis of atomically dispersed Mn-N-C catalysts from aqueous solution [98]. First, Mn-doped ZIF-8 precursor was synthesized in HCl aqueous solution, and then the Mn-doped ZIF-8 was carbonized at high temperature to evaporate Zn and create a porous carbon host structure, and MnN_4_ sites were created by high temperature. Then, the step pyrolysis strategy (800 °C/1100 °C) was used to adsorb Mn ions on the Mn-N-C-first catalyst to significantly increase the density of the active sites in the micropores. Finally, the Mn-N-C catalyst with high activity and strong durability was obtained through the second thermal activation. The Mn-N-C catalyst synthesis was shown in Figure 8a. The images of Mn-N-C-HCl-800/1100-first catalyst and Mn-N-C-HCl-800/1100 catalyst were obtained by high-angle annular dark-field scanning transmission electron microscopy (HAADF-STEM) method as shown in Figure 8b,c, which showed the curved-surface polyhedron morphology of carbon particles. As shown in Figure 8d, the Mn-N-C-HCl-800/1100 catalyst exhibited high activity with an E_1/2_ of 0.815 V (vs. RHE). And E_1/2_ exhibited excellent stability with a loss of only 14 mV after 30,000 cycles (see Figure 8e).

#### 2.1.4. Other SACs

Due to the nearly 100% utilization of metal atoms, the strong metal- support interaction, and the low coordination environment of SACs [99], other single atoms besides Fe, Mn, Co can also be the metal center atoms of SACs, such as Cu, Ni and Zn. The performance of other metal SACs, including activity and stability, are reviewed in detail in Table 1. Other heteroatoms, such as S, F, and N co-doped metal atoms, are also taken into account to show the progress of SACs more comprehensively.

### 2.2. Double-Atom Catalysts (DACs)

Although researchers have made great progress in the research and synthesis of single atom catalysts, the activity and stability of SACs are still difficult to reach the best state, mainly because of the inherent electronic structure of single metal atoms, which hinders the effective development of catalyst activity [106,107]. By introducing other metal atoms to synthesize double atom catalysts (DACs), which have the advantages of high utilization and two metal atom sites, it is considered as a promising catalyst for the ORR of PEMFCs [108,109]. The synthesis of DACs can change the properties of each metal, and improve their intrinsic performance to achieve high activity and durability. In particular, the synthesis of bimetallic catalysts from N-coordinated bimetallic atoms has become a hot topic of research, and some DACs, such as FeCo-DACs [110,111], FeMn-DACs [112,113], MnCo-DACs [114,115], have been studied for the ORR of PEMFCs. However, the catalytic mechanism of DACs has not been accurately determined at present [116].

#### 2.2.1. FeCo-DACs

Generally, single metal atoms Fe and Co have high catalytic activity for oxygen reduction reaction (ORR) and oxygen evolution reaction (OER), respectively [117]. After combining Fe and Co to prepare FeCo-DACs, the activity of catalyzing ORR can be greatly enhanced [118]. Some papers have made progress in the preparation and research of FeCo DACs. Wu et al. reported a ZIF-derived FeCo-N co-doped carbon nanoframework (FeCo-NC) [119]. The synthesis process and the structure of FeCo-NC can be seen from Figure 9a. Concretely, Zn(NO_3_)_2_, Co(NO_3_)_2_ and 2-MIM were heated in methanol solution for 4 h to assemble Co/Zn ZIF firstly. And then Fe(acac)_3_ was trapped in the cavity to obtain Fe/Co/Zn ZIF. Finally, the catalyst was carbonized at 900 °C for 3 h to obtain FeCo–NC catalyst. Samad et al. used thermal annealing to obtain a FeCo/NG catalyst consisting of iron and cobalt (Fe and Co) double atoms supported on N-doped graphene (see Figure 9b) [120]. Specifically, graphene oxide(GO), dicyandiamide (DCDA), FeCl_3_ and Co(NO_3_)_2_·6H_2_O were used as the precursors of O, N, Fe and Co respectively, and DCDA is added to the aqueous solution of GO. And after ultrasonic treatment for 2 h, FeCl_3_ and Co(NO_3_)_2_·6H_2_O were added, and then the mixture was stirred continuously at 80 °C for 24 h. Finally, the mixed powder was annealed at high temperature (600–800 °C) in N_2_ atmosphere for 2 h to obtain a:b-FeCo/NG-n (a:b is the molar ratio, n is the annealing temperature). Chen et al. reported a method for the synthesis of a FeCo double atoms and N co-doped C catalysts using ZIF-8 precursor [121]. As shown in Figure 9c, ZIF-8 was prepared by simply mixing Zn(NO_3_)_2_·6H_2_O and 2-MIM in methanol firstly. And then, Fe(NO_3_)_3_·9H_2_O and Co(NO_3_)_2_·6H_2_O were added and reacted with ZIF-8 to form FeCo/ZIF-8. Finally, with the assistance of NaCl salt, FeCo/ZIF-8 was carbonized and unfolded to synthesize ultrathin Fe, Co, N-codoped graphite flake (FeCo/NG), while the pyrolysis of FeCo/ZIF-8 without NaCl yields Fe, Co, N-codoped carbon spheres (FeCo/NC).

At present, the ORR activity of FeCo-N-C DACs prepared by Fe, Co and N co-doping C materials have exceeded that of commercial Pt/C catalysts (20 wt% of Pt, Johnson Matthey) in alkaline electrolyte and is equivalent to that of Pt/C catalyst in acidic electrolyte [122,123]. To further prove the activity and stability of FeCo-DACs, Im et al. used 2D ZIF as the core and 3D ZIF as the shell, synthesized core-shell-type leaf-shaped CoFe-NC catalysts [124]. The synthesis process was shown in Figure 10a, and transmission electron microscope (TEM) image of L-CoFe-NC and the element mapping images of cobalt and iron are shown in Figure 10b–d respectively, showing that Co and Fe are uniformly distributed in the C frame. By adjusting the concentration of Fe doping, the ORR activity of CoFe-NC catalysts was obtained. As shown in Figure 10e, when the ratio of Fe was 0.5 (CoFe_0.5_-NC), the ORR activity was the best, and the half-wave potential was 0.77 V. At the same time, CoFe_0.5_-NC exhibited excellent durability with almost the same LSV curve even after 10,000 ADT cycles (see Figure 10f). Finally, L-CoFe_0.5_-NC catalyst was used in PEMFC, and the PEMFC exhibited an open circuit voltage of 0.731 V and a maximum power density of 145 mw/cm^2^ (see Figure 10g).

#### 2.2.2. FeMn-DACs

The introduction of a second metal atom can regulate the electronic structure of the Fe-N site and effectively improve the catalytic activity of ORR [125,126]. Some papers chose Mn as the second metal atom to transform Fe-N-C SAC into FeMn-N-C DAC [127,128]. Huang et al. synthesized a Fe-Mn-N-C DAC with new local structure of FeN_4_-MnN_3_ [129], the synthesis routes were shown in Figure 11a. Concretely, Zn, Mn metal salts and 2-MIM were stirred and assembled to obtain Mn ZIF precursor firstly. And then, Mn ZIF precursor was pyrolyzed in N_2_ atmosphere to obtain Mn-N-C precursor. Finally, the Mn-N-C precursor was adsorbed with Fe and N sources by the double solvent method, and the Fe-Mn-N-C catalyst was obtained by the second pyrolysis. The test results showed that the presence of Fe, Mn double sites increased the catalytic activity of Fe-Mn-N-C. As shown in Figure 11b,c, the Fe-Mn-N-C catalyst exhibited a half-wave potential of 0.79 V (vs. RHE) in 0.1 M HClO_4_ solution, which was slightly weaker than the commercial 20 wt% Pt/C catalyst; And the E_1/2_ of Fe-Mn-N-C catalyst in 0.1 M KOH solution achieved 0.93 V (vs. RHE), which was higher than 20 wt% Pt/C. And in Figure 11d, the power peak density of Fe-Mn-N-C-based PEMFCs achieved 1.048 W/cm^2^, indicating a good practical application prospect.

#### 2.2.3. MnCo-DACs

In order to completely avoid Fenton reaction and improve ORR performance, Fe-free DACs have become a hotspot [130,131]. Considering the higher selectivity of Mn for the four-electron ORR pathway and the high activity of Co, the activity and selectivity problems can be solved by combining Mn with Co [132,133]. Zhang et al. reported a method for synthesizing Mn/CO DACs [134], in which manganese and cobalt salts were used as metal precursors and urea was used as carbon source and nitrogen source to synthesize the Mn/Co-bamboo-like N-doped carbon nanotubes (Mn/Co-BNCNTs) catalyst. The synthesis method of Mn/Co-BNCNTs catalyst is facile, practicable and reproducible (Figure 12). Hou et al. synthesized bimetallic catalysts by the same method [135].

Heteroatom-doped carbon nanotubes (CNTs) have become a popular choice for the synthesis of metal catalysts due to their large surface area and large aspect ratio [90,134,135,136,137]. However, it is difficult to obtain catalysts with homogeneous distribution of metal atoms and to exert their optimal catalytic performance [138]. MOFs once again become attractive potential precursors for the synthesis of DACs. Shah et al. reported a facile and controlled sacrificial-template synthesis method by using ZIF-8 precursor to prepare MnCo-NC/CNT catalyst [139], and the synthesis procedure of MnCo-NC/CNT was shown in Figure 13a. In the process of preparation, the ratio of Co to Mn was controlled at 2:1 and mixed with ZIF-8 solution to prepare MnCo-ZIF-8 polyhedron. Then MnCo-NC/CNT catalyst was prepared by two successive pyrolysis steps (550 °C@4 h and 900 °C@3 h). As shown in Figure 13b–e, MnCo-NC/CNT catalyst in acidic (0.1 M HClO_4_) and alkaline (0.1 M KOH) electrolytes had a half-wave potential of 0.83 V and 0.90 V, respectively, showing excellent ORR performance. After long-term durability test, MnCo-NC/CNT catalyst showed better stability than Pt/C catalyst.

#### 2.2.4. Other DACs

Due to the variety of transition metals, there is a great space for the synthesis of double metal atoms catalysts [140,141]. In addition to FeCo-DACs, FeMn-DACs and MnCo DACs, there are other DACs synthesized by two other different TM atoms, such as FeCu-DACs, FeNi-DACs, FeZn-DACs, CuZn-DACs. At the same time, DACs with other non-metallic elements (such as S, P and O) replacing N or co-doping carbon substrate with N are also synthesized. The performance of other metal DACs, including activity and stability, are reviewed in detail in Table 2.

The ORR durability of non-noble metal catalysts is of great value for real commercial applications. However, the above studies only tested ORR durability in a laboratory environment and did not consider measures to improve the ORR durability of non-noble metal catalysts. Atomic scale metal elements have high surface energy, which cause single metal atoms to tend to aggregate and destroy the stability of SACs [150,151]. For DACs, the introduction of metal atoms in a different d-band can effectively adjust the electronic structure and improve the ORR durability of the catalysts [152,153]. Therefore, studies on the durability of non-noble metal catalysts mainly focus on the improvement of the stability of SACs [154,155,156]. Wang et al. concluded that defect-anchoring strategies and confinement strategies were the two most common stability strategies; these strategies can enhance the interaction between metal atoms and the support [157,158]. For example, Abdul Majid et al. reported that single Cu atoms anchoring and capping defect sites on the Zr oxide clusters of UiO-66 could improve the stability of Cu/UiO-66 catalysts [159]. The effective interaction between single metal atoms and the support can not only prevent clusters between atoms, but also regulate the electronic structure of the catalysts [160]. Therefore, the surface and microstructure of the support are the key factors for improving the stability of the SACs; these factors are relatively easy to control.

## 3. Conclusions and Perspectives

In order to improve the output power, dynamic response, life and other comprehensive performance aspects of PEMFCs, thereby accelerating their commercial process, it is urgent and meaningful to explore efficient and durable non-noble metal ORR catalysts. This paper mainly reviewed the research on non-noble metal ORR catalysts for PEMFCs in the past five years from the perspective of preparation and performance, which mainly included two categories: single transition metal atom catalysts and double transition metal atom catalysts. Generally, there are two main methods used to synthesize non-noble metal ORR catalysts: (1) mixing and direct pyrolysis; (2) a sacrificial-template method based on MOF, followed by pyrolysis to obtain the catalyst. The precursor type, precursor structure, heat treatment time, heat treatment temperature and post-treatment operation of the preparation method will have a significant impact on the activity and stability of the non-noble metal ORR catalysts. Furthermore, the surface area, active site and exposure rate of non-noble metal ORR catalysts directly affect the catalytic activity and stability. Therefore, in the design and preparation process of non-noble metal catalysts, it is important to select promising precursors, strictly control the heat treatment and post-treatment conditions, and strive to improve the surface area, active site and exposure rate. Although great progress has been made in the preparation and performance of non-noble metal ORR catalysts for PEMFCs, there are still many challenges.

Firstly, there are many methods available to synthesize SACs and DACs, but reducing the cost, shortening the synthesis cycle and improving the practicability of preparation methods is still a challenge. Secondly, regarding the preparation of SACs and DACs, methods to precisely control the synthesis conditions and obtain catalysts with high surface area, multiple active sites and exposure require further study. Additionally, the existing research on SACs has mainly focused on Fe, Mn, Co and Cu atoms, and the DACs mostly consisted of the above atoms, as well. Thus, the influence of the introduction of other transition metal atoms on the performance of TM–N–C catalysts still needs further exploration. Moreover, the selection and use of C carriers can be further optimized or replaced to synthesize highly active and durable catalysts. Finally, it is difficult for most catalysts to exceed the overall performance of Pt/C catalysts in acidic conditions; therefore, improving the activity and durability of catalysts in acidic conditions is still a challenge.

Undoubtedly, advanced non-noble metal catalysts have exhibited excellent activity and durability, showing similar performance to commercial Pt/C catalysts for fuel cell applications; some non-noble metal catalysts even outperform the DOE’s 2020 performance targets. Notably, the durability test of non-noble metal catalysts was carried out under laboratory conditions, which is still very different from the actual complex and changeable application scenarios. Improving the durability of non-noble metal catalysts is still a challenge for the commercial application of fuel cells. In the future, the design and preparation for ORR catalysts of PEMFCs should follow the comprehensive objectives of high activity, high durability, low price and scalability, and further optimize the preparation method to guide the realization of large-scale production and application of efficient and durable catalysts at an early date.

## Figures and Tables

**Figure 1 nanomaterials-12-03331-f001:**
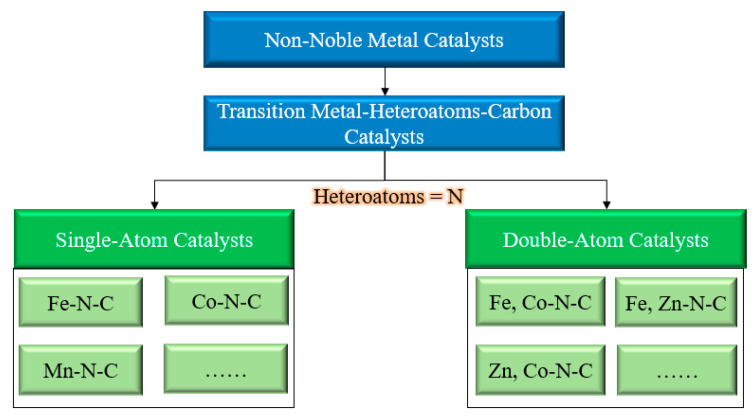
Scope and boundary of review.

**Figure 2 nanomaterials-12-03331-f002:**
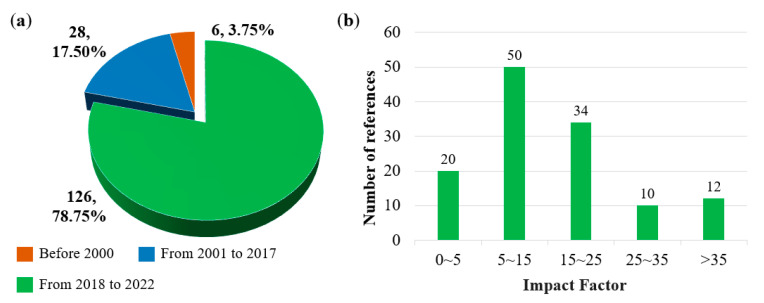
Analysis of references: (**a**) Classified by year; (**b**) Impact factor in last five years.

**Figure 4 nanomaterials-12-03331-f004:**
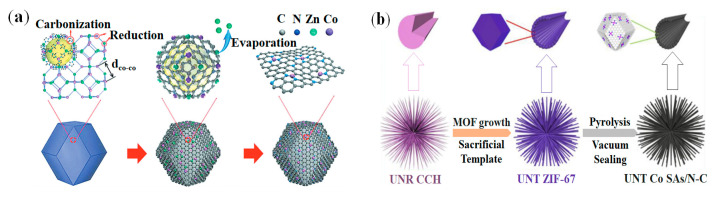
Synthesis methods of Co-SACs: (**a**) Synthesis from bimetallic Zn/Co MOF. Adapted with permission from Ref. [75]. Copyright 2016 John Wiley and Sons. (**b**) Synthesis from UNR CCH. Adapted with permission from Ref. [81]. Copyright 2019 Elsevier.

**Figure 5 nanomaterials-12-03331-f005:**
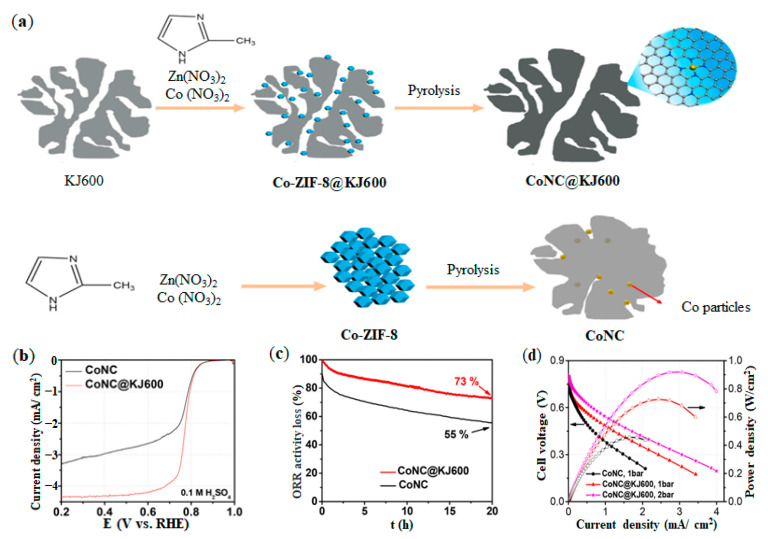
Synthesis method and performance of CoNC@KJ600 and CoNC catalysts: (**a**) Synthesis diagram; (**b**) ORR activity; (**c**) ORR durability; (**d**) PEMFC performance. Adapted with permission from Ref. [84]. Copyright 2020 Elsevier.

**Figure 6 nanomaterials-12-03331-f006:**
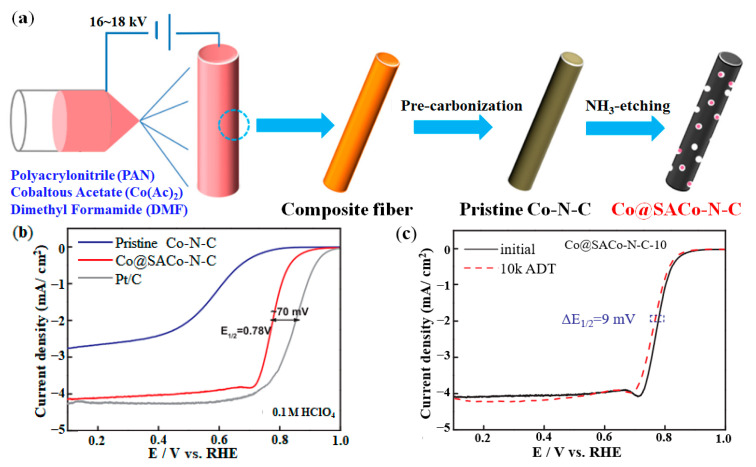
Synthesis method and performance of Co@SACo-N-C catalyst: (**a**) Synthesis diagram; (**b**) ORR activity; (**c**) ORR durability. Adapted with permission from Ref. [89]. Copyright 2018 Elsevier.

**Figure 7 nanomaterials-12-03331-f007:**
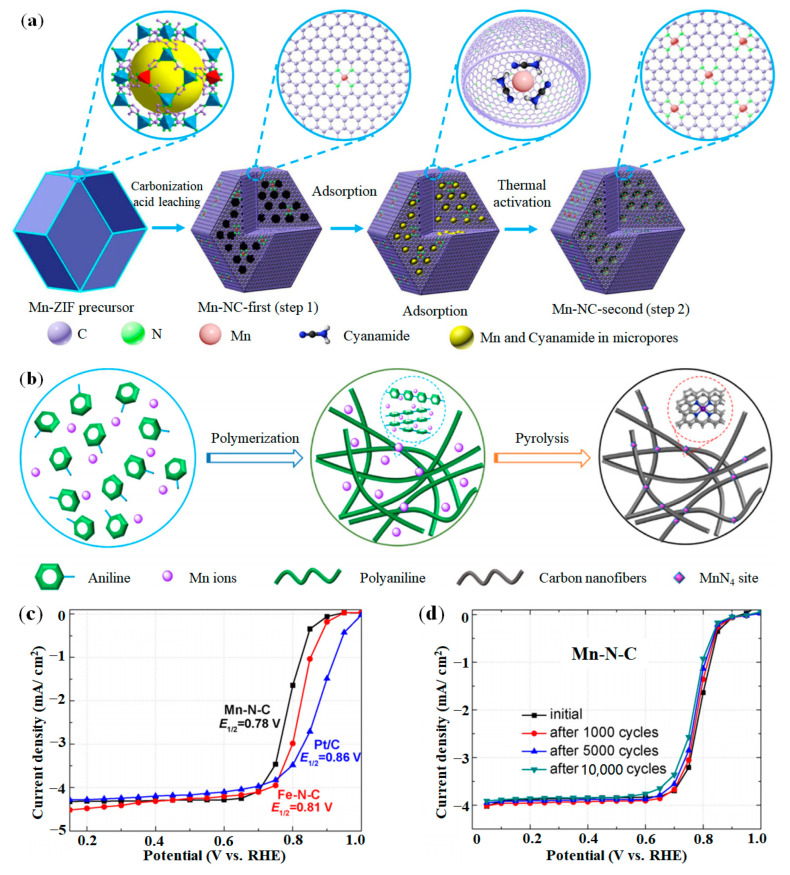
Synthesis methods and performance of Mn-SAC: (**a**) Synthesis strategy of Mn-NC catalyst. Adapted with permission from Ref. [90]. Copyright 2018 springer nature. (**b**) Synthesis strategy of Mn-N-C catalyst; (**c**) ORR activity; (**d**) ORR durability. Adapted with permission from Ref. [97]. Copyright 2019 Elsevier.

**Figure 8 nanomaterials-12-03331-f008:**
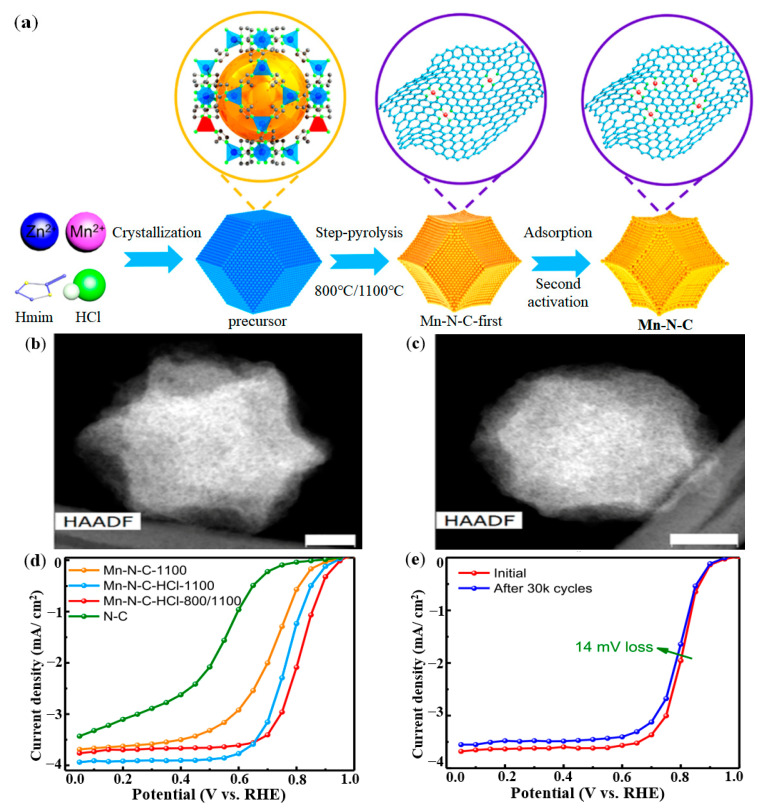
Synthesis method and performance of Mn-N-C catalyst: (**a**) Synthesis strategy of Mn-N-C catalyst; (**b**) HAADF-STEM image of Mn-N-C-HCl-800/1100-first catalyst; (**c**) HAADF-STEM image of Mn-N-C-HCl-800/1100 catalyst; (**d**) ORR activity; (**e**) ORR durability. Adapted with permission from Ref. [98]. Copyright 2020 American Chemical Society.

**Figure 9 nanomaterials-12-03331-f009:**
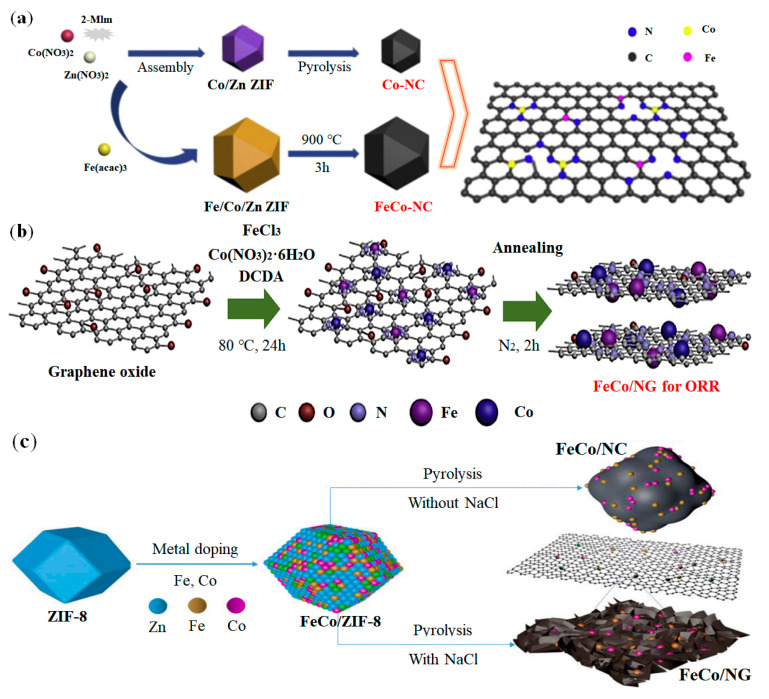
Synthesis methods of FeCo-DACs: (**a**) Synthesis strategy of FeCo-NC catalyst. Adapted with permission from Ref. [119]. Copyright 2018 Elsevier. (**b**) Synthesis strategy of FeCo-NG catalyst. Adapted with permission from Ref. [120]. Copyright 2020 Elsevier. (**c**) Synthesis strategy of FeCo-NC and FeCo-NG catalysts. Adapted with permission from Ref. [121]. Copyright 2022 Elsevier.

**Figure 10 nanomaterials-12-03331-f010:**
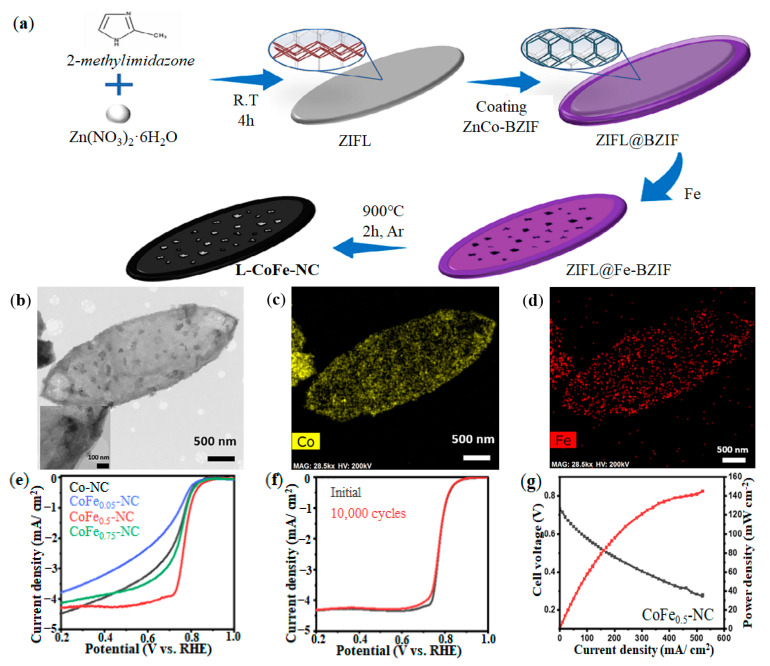
Synthesis method and performance of L-CoFe-NC catalyst: (**a**) Synthesis process of L-CoFe-NC catalyst; (**b**) TEM image of L-CoFe-NC; (**c**) Elemental mapping image of Co; (**d**) Elemental mapping image of Fe; (**e**) ORR activity; (**f**) ORR durability; (**g**) PEMFC performance. Adapted with permission from Ref. [124]. Copyright 2021 John Wiley and Sons.

**Figure 11 nanomaterials-12-03331-f011:**
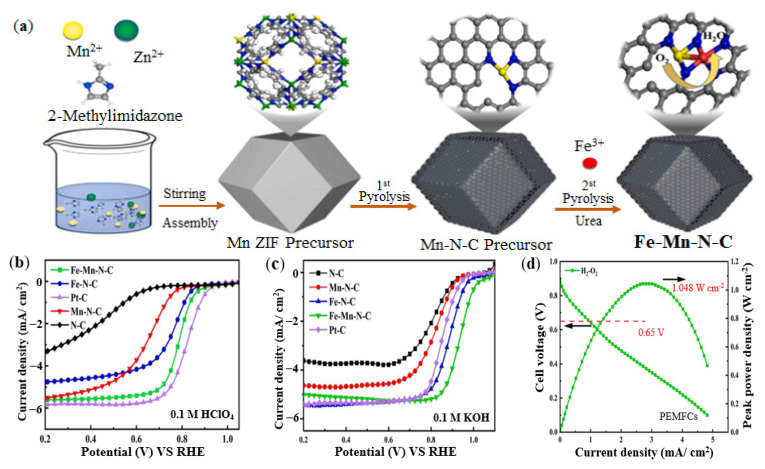
Synthesis method and performance of Fe-Mn-N-C catalyst: (**a**) Synthesis routes of Fe-Mn-N-C catalyst; (**b**) ORR activity in 0.1 M HClO_4_; (**c**) ORR activity in 0.1 M KOH; (**d**) PEMFC performance. Adapted with permission from Ref. [129]. Copyright 2022 Elsevier.

**Figure 12 nanomaterials-12-03331-f012:**
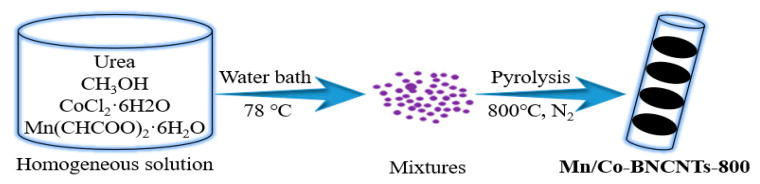
The synthesis method of Mn/Co-BNCTs catalyst. Adapted with permission from Ref. [134]. Copyright 2019 Elsevier.

**Figure 13 nanomaterials-12-03331-f013:**
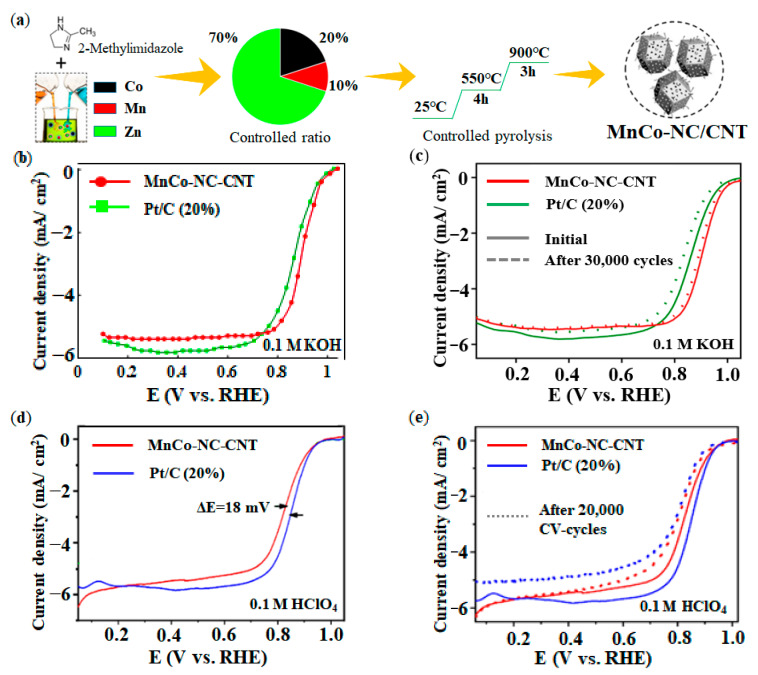
Synthesis method and performance of MnCo-NC/CNT catalyst: (**a**) Synthesis procedure of MnCo-NC/CNT catalyst; (**b**) ORR activity in 0.1 M KOH; (**c**) ORR durability in 0.1 M KOH; (**d**) ORR activity in 0.1 M HClO_4_; (**e**) ORR durability in 0.1 M HClO_4_. Adapted with permission from Ref. [139]. Copyright 2021 American Chemical Society.

**Table 1 nanomaterials-12-03331-t001:** Summary of the ORR performance for other metal SACs.

Catalysts	Specific Name	E_1/2_/Acid Electrolyte	Stability	References
Cu-SACs	Cu_SA_/Cu_CT_@NPC	0.80 V vs. RHE/0.1 M HClO_4_	10,000 cycles/6 mV negative shift	[100]
	Cu-SAs/NSs	0.74 V vs. RHE/0.1 M HClO_4_	3000 cycles	[101]
Ni-SACs	Ni-N_3_-Gra	Comparable with Pt	-	[102]
Zn-SACs	Zn-N-C	0.746 V vs. RHE/0.1 M HClO_4_	1000 cycles/19.88 mV negative shift	[103]
	Zn-B/N-C	0.753 V vs. RHE/0.1 M HClO_4_	Current density/mA·cm^−2^ (87% remained after 80,000 s)	[104]
	A-Zn@NSG	0.805 V vs. RHE/0.1 M HClO_4_	5000 cycles/6.7 mV negative shift	[105]

**Table 2 nanomaterials-12-03331-t002:** Summary of the ORR performance for other metal DACs.

Catalysts	Specific Name	E_1/2_/Acid Electrolyte	Stability	References
FeCu-DACs	FeCu-N-C	0.784 V vs. RHE/0.1 M HClO_4_	10,000 cycles/15 mV negative shift	[142]
	FeCu/N-CNTs	0.811 V vs. RHE/0.1 M HClO_4_	5000 cycles	[143]
	FeCuNC	0.820 V vs. RHE/0.5 M H_2_SO_4_	-	[144]
FeNi-DACs	FeNi-N_6_-C	0.780 V vs. RHE/0.1 M HClO_4_	5000 cycles/Almost unchanged	[145]
	Fe/Ni-N_X_/OC	0.840 V vs. RHE/0.1 M HClO_4_	5000 cycles/Almost unchanged	[146]
FeZn-DACs	Fe-Zn-SA/NC	0.780 V vs. RHE/0.1 M HClO_4_	5000 cycles/No obvious decrease	[147]
	Zn/Fe–N–C	0.810 V vs. RHE/0.5 M H_2_SO_4_	40,000 s/85.6% current retained	[148]
CoNi-DACs	CoPNi-N/C	0.730 V vs. RHE/0.1 M HClO_4_	5000 cycles/11 mV negative shift	[149]

## Data Availability

Not applicable.

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
