# Peer review of "Non-Noble Metal Catalysts in Cathodic Oxygen Reduction Reaction of Proton Exchange Membrane Fuel Cells: Recent Advances"

_nanomaterials, 2022, doi:10.3390/nano12193331_

Round 1
Reviewer 1 Report
In this manuscript, authors reviewed the Non-Noble Metal Catalytic materials for Cathodic Oxygen Reduction Reaction of Proton Exchange Membrane Fuel Cells. The approach on non-noble metal catalysts is not very new and but could be informative to some readers. However, there is a lack of the construction of the main body to properly understand the trend on non-noble metal catalysts. Therefore, I do not recommend this manuscript to be published unless some changes are made.
There are some comments
1. I am not quite sure why this was submitted to Nanomaterials. I think it is more related to Catalysts or Energies. Also, authors focused on the catalytic compositions and synthesis methods not the nano morphology such as nanoparticle or pore physical structure, size, and shape etc. These can be improved.
2. The review should introduce what are the important things when the non-noble metal catalysts are designed.
3. The review is mainly introducing metals of Fe, Mn and Co. The author should explain why these metals are effective as non-noble metal catalysts.
4. The positions of the Figures are all different. Please modify.
5. The review should summarize the performance difference of they want to say that it is still challenging. Maybe, the target performance has to be summarize to compare the performances. For example, DOE has reported the performance targets for activity and durability of the non-noble metal catalysts.
6. The overall English need to be improved.
Author Response
Dear Reviewer,
We would like to express our sincere appreciation for your careful reading and invaluable comments to improve this paper. In order to effectively improve the quality of the paper, we have made the following modifications to your comments. The amendments made are mentioned below with reference to appropriate paragraphs and sections of the revised manuscript.
[Comment 1] I am not quite sure why this was submitted to Nanomaterials. I think it is more related to Catalysts or Energies. Also, authors focused on the catalytic compositions and synthesis methods not the nano morphology such as nanoparticle or pore physical structure, size, and shape etc. These can be improved.
[Response] Thank you for your careful comment. The manuscript was submitted to the topic of Catalysis for Sustainable Chemistry and Energy of Nanomaterials. And review papers dealing with all types of catalysis, including electrocatalysis, fall within the scope of this Topic Issue. Also, we thank you for your comment on nanoparticle or pore physical structure, size, and shape. Introduction of pore physical structure, size, and shape of catalysts will be added to the manuscript.
[Comment 2] The review should introduce what are the important things when the non-noble metal catalysts are designed.
[Response] Thanks for your valuable suggestions. And we introduce the important things when the non-noble metal catalysts are designed in Section 3 on page 15.
The precursor type, precursor structure, heat treatment time, heat treatment temperature and post-treatment operation in the preparation method will have an important impact on the activity and stability of the non-noble metal ORR catalysts. Furthermore, the surface area, active site and exposure rate of non-noble metal ORR catalysts directly affect the catalytic activity and stability. Therefore, in the design and preparation process of non-noble metal catalysts, it is important to select promising precursors, strictly control the heat treatment and post-treatment conditions, and strive to improve the surface area, active site and exposure rate of catalysts.
[Comment 3] The review is mainly introducing metals of Fe, Mn and Co. The author should explain why these metals are effective as non-noble metal catalysts.
[Response] Thanks for your precious advice. And we will explain the metals of Fe, Mn and Co are effective as non-noble metal catalysts in Section 1 on page 2.
Due to the abundant reserves, low price and strong scalability, Fe, Co and Mn have been valued by researchers [1]. Recently, various transition metal-nitrogen-carbon catalysts (TM-N-C catalysts, TM: Fe, Mn, Co, etc.) have been studied and prepared, and they have shown promising electrocatalytic activity and durability [2,3]. The main reason for the excellent performance of TM-N-C catalysts is the synergistic effect between transition metal atoms, nitrogen, and carbon materials [4]. Furthermore, with the help of spectroscopy technique and density functional theory (DFT), it was found that the active sites of atomic metal coordinated nitrogen sites (such as, Fe-NX, Co-NX and Mn-NX.) was the main reason leading to the activity of TM-N-C catalysts [5-7]. However, the structure of TM-N-C active site is complex and may be dynamically changed during ORR process, so it is a challenge to clearly describe the reasons for the improved ORR performance [8].
[1] He, Y.H.; Liu, S.W.; Shi, Q.R.; Wu, G. Atomically dispersed metal-nitrogen-carbon catalysts for fuel cells: advances in catalyst design, electrode performance, and durability improvement. Chem. Soc. Rev. 2020, 49(11), 3484-3524.
[2] Shen, M.X.; Wei, C.T.; Ai, K.L.; Lu, L.H. Transition metal-nitrogen-carbon nanostructured catalysts for the oxygen reduction reaction: From mechanistic insights to structural optimization. Nano Research 2017, 10(5), 1449-1470.
[3] Kiani, M.; Zhang, J.; Luo, Y.; Jiang, C.P.; Fan, J.L.; Wang, G.; Chen, J.W.; Wang, R.L. Recent developments in electrocatalysts and future prospects for oxygen reduction reaction in polymer electrolyte membrane fuel cells. J. Energy Chem. 2018, 27(4), 1124-1139.
[4] Wang, S.C.; Teng, Z.Y.; Wang, C.Y.; Wang, G.X. Stable and Efficient Nitrogen-Containing Carbon-Based Electrocatalysts for Reactions in Energy-Conversion Systems. ChemSusChem 2018, 11(14), 2267-2295.
[5] Yang, H.; Chen, X.; Chen, W.T.; Wang, C.; Cuello, N.C.; Nafady, A.; Al-Enizi, A.M.; Waterhouse, G.I.N.; Goenaga, G.A.; Zawodzinski, T.A.; Kruger, P.E.; Clements, J.E.; Zhang, J.; Tian, H.; Telfer, S.G.; Ma, S.Q. Tunable Synthesis of Hollow Metal-Nitrogen-Carbon Capsules for Efficient Oxygen Reduction Catalysis in Proton Exchange Membrane Fuel Cells.
[6] Hu, B.; Zhu, X.B.; An, X.H.; Wang, C.X.; Wang, X.B.; He, J.L.; Zhao, Y. Separation of Metal-N4 Units in Metal-Organic Framework for Preparation of M-Nx/C Catalyst with Dense Metal Sites. Inorg. Chem. 2020, 59(23), 17134-17142.
[7] Luo, E.G.; Chu, Y.Y.; Liu, J.; Shi, Z.P.; Zhu, S.Y.; Gong, L.Y.; Ge, J.J.; Choi, C.H.; Liu, C.P.; Xing, W. Pyrolyzed M-NX catalysts for oxygen reduction reaction: progress and prospects. Energy Environ. Sci. 2021, 14(4), 2158-2185.
[8] Deng, Y.J.; Luo, J.M.; Chi, B.; Tang, H.B.; Li, J.; Qiao, X.C.; Shen, Y.J.; Yang, Y.J.; Jia, C.M.; Rao, P.; Liao, S.J.; Tian, X.L. Advanced Atomically Dispersed Metal-Nitrogen-Carbon Catalysts Toward Cathodic Oxygen Reduction in PEM Fuel Cells. Adv. Energy Mater. 2021, 11(37), 2101222.
[9] Mohideen, M.M.; Radhamani, A.V.; Ramakrishna, S.; Wei, Y.; Liu, Y. Recent insights on iron based nanostructured electrocatalyst and current status of proton exchange membrane fuel cell for sustainable transport. J. Energy Chem. 2022, 69, 466-489.
[Comment 4] The positions of the Figures are all different. Please modify.
[Response] Thanks for your rigorous advice. We will carefully check the positions of the Figures in the document and modify them.
[Comment 5] The review should summarize the performance difference of they want to say that it is still challenging. Maybe, the target performance has to be summarize to compare the performances. For example, DOE has reported the performance targets for activity and durability of the non-noble metal catalysts.
[Response] Thanks for your detailed suggestions. We summarize the performance targets of non-noble metal catalysts reported by the DOE in Section 1 on page 1-2.
According to the US Department of Energy (DOE), noble metal catalysts account for almost 60% of the cost of fuel cell systems, which has greatly hindered the commercial application of fuel cells [1]. Using non-noble metals to replace Pt to design and prepare non-noble metal catalysts has become a promising measure to reduce costs. In order to effectively overcome the cost and durability challenge of fuel cell electrocatalysts, the US DOE has set the performance target of activity and durability of non-noble metal catalysts. Specifically, the US DOE set the 2020 activity target for non-noble group metal catalyst as 0.044 A/cm2 at 0.9 ViR-free under 1 bar H2-O2 [2], and the 2020 target for membrane electrode durability should be more than 5000 h, with no more than 30 mV of performance loss while minimizing cost and meeting the target durability [3,4]. In recent years, the researchers aim to make the performance of the designed and prepared non-noble metal catalysts close to or exceed the target performance of DOE. The non-noble metal catalysts have made great progress in improving the cathodic ORR activity and durability of PEMFCs, and several review papers have been published to evaluate the progress of non-noble metal catalysts [5-9].
[1] Mohideen, M.M.; Liu, Y.; Ramakrishna, S. Recent progress of carbon dots and carbon nanotubes applied in oxygen reduction reaction of fuel cell for transportation. Appl. Energy 2020, 257, 114027.
[2] He, Y.H.; Wu, G. PGM-Free Oxygen-Reduction Catalyst Development for Proton-Exchange Membrane Fuel Cells: Challenges, Solutions, and Promises. Acc. Mater. Res. 2022, 3(2), 224-236.
[3] Wang, Y.; Yuan, H.; Martinez, A.; Hong, P.; Xu, Hui.; Bockmiller, F.R. Polymer electrolyte membrane fuel cell and hydrogen station networks for automobiles: Status, technology, and perspectives. Adv. Appl. Energy 2021, 2, 100011.
[4] Chen, M.J.; Li, C.Z.; Zhang, B.Z.; Zeng, Y.C.; Karakalos, S.; Hwang, S.; Xie, J.; Wu, G. High-Platinum-Content Catalysts on Atomically Dispersed and Nitrogen Coordinated Single Manganese Site Carbons for Heavy-Duty Fuel Cells. J. Electrochem. Soc. 2022, 169(3), 034510.
[5] Zeng, K.; Zheng, X.J.; Li, C.; Yan, J.; Tian, J.H.; Jin, C.; Strasser, P.; Yang, R.Z. Recent Advances in Non-Noble Bifunctional Oxygen Electrocatalysts toward Large-Scale Production. Adv. Funct. Mater. 2020, 30(27), 2000503.
[6] Tang, T.; Ding, L.; Jiang, Z.; Hu, J.S.; Wan, L.J. Advanced transition metal/nitrogen/carbon-based electrocatalysts for fuel cell applications. Sci China Chem 2020, 63(11), 1517-1542.
[7] Deng, Y.J.; Luo, J.M.; Chi, B.; Tang, H.B.; Li, J.; Qiao, X.C.; Shen, Y.J.; Yang, Y.J.; Jia, C.M.; Rao, P.; Liao, S.J.; Tian, X.L. Advanced Atomically Dispersed Metal-Nitrogen-Carbon Catalysts Toward Cathodic Oxygen Reduction in PEM Fuel Cells. Adv. Energy Mater. 2021, 11(37), 2101222.
[8] Ye, C.W.; Xu, L. Recent advances in the design of a high performance metal-nitrogen-carbon catalyst for the oxygen reduction reaction. J. Mater. Chem. A 2021, 9(39), 22218-22247.
[9] Lian, J.; Zhao, J.Y.; Wang, X.M. Recent Progress in Carbon-based Materials of Non-Noble Metal Catalysts for ORR in Acidic Environment. Acta Metall Sin-Engl 2021, 34(7), 885-899.
And we also summarize the challenge between the performance of non-noble metal catalysts and DOE performance targets in Section 3 on page 15.
Undoubtedly, advanced non-noble metal catalysts have exhibited excellent activity and durability, showing similar performance to commercial Pt/C catalysts for fuel cell applications, and some non-noble metal catalysts even outperform 2020 target performance of DOE. Notably, the durability test of non-noble metal catalysts is carried out under laboratory conditions, which is still very different from the actual complex and changeable application scenarios. The durability of non-noble metal catalysts is still a challenge for commercial application of fuel cells. In the future, the design and preparation for ORR catalysts of PEMFCs should follow the comprehensive objectives of high activity, high durability, low price and scalability, and further optimize the preparation method to guide the realization of large-scale production and application of efficient and durable catalysts at an early date.
[Comment 6] The overall English need to be improved.
[Response] Thanks for your professional advice. And we read the manuscript carefully and revise it to improve the overall English level, and the modifications are marked in green.
Once again, we would like to express our most sincere thanks for your careful reading and valuable comments.

Reviewer 2 Report
1. Many instances of incorrect English usage render the manuscript with a serious deficiency in comprehending the author's statements. For eg., a. Introduction section: Page 1 Ln: 40-41: Incomplete sentence with grammatical error; b. Page: 2 Ln: 55-56, 62 - Incorrect sentence, c. Page: 3 Ln: 88-89 - Incorrect sentence
2. I have seen many review articles published on the topic of TM-N-C type ORR catalysts applied in PEFC cathodes. Why should another review be published on this topic? This has to be clearly described in the introduction section with emphasis on the points that are distinctly covered in the review and not in other similar reviews published earlier.
3. The statement in the abstract and introduction section refers only to the PEMFC catalyst. However, in Table 1, the authors present the activity of the catalyst measured in alkaline pH. How this is relevant to PEMFC? The same comment holds true for Table 2. If the review is not exclusively focused on PEMFC, the statements in the appropriate sections using the acronym PEMFC should be deleted or modified.
4. Conclusion section lacks clarity.
Based on the above comments, I cannot recommend publication before the authors have adequately addressed my concerns.
Author Response
Dear Reviewer,
We would like to express our sincere appreciation for your careful reading and invaluable comments to improve this paper. In order to effectively improve the quality of the paper, we have made the following modifications to your comments. The amendments made are mentioned below with reference to appropriate paragraphs and sections of the revised manuscript.
[Comment 1] Many instances of incorrect English usage render the manuscript with a serious deficiency in comprehending the author's statements. For eg., a. Introduction section: Page 1 Ln: 40-41: Incomplete sentence with grammatical error; b. Page: 2 Ln: 55-56, 62 - Incorrect sentence, c. Page: 3 Ln: 88-89 - Incorrect sentence
[Response] Thank you for your professional and valuable advice. We have carefully revised our manuscript.
# Page 1 Ln: 40-41: In this paper, the non-noble metal catalyst accurately identified as transition metal-heteroatoms-carbon catalysts (TM-H-C catalysts, TM: Fe, Mn, Co, etc.).
# In this paper, the non-noble metal catalysts were accurately identified as transition metal-heteroatoms-carbon catalysts (TM-H-C catalysts).
# Page: 2 Ln: 55-56: Extensive and high impact factor references, indicating the advanced, scientific and instructive of this paper.
# Thank you for pointing out the incorrect sentence and we deleted the sentence.
# Page: 2 Ln: 62: This paper has guiding significance and reference value for the progress of non-noble metal catalysts.
# Thank you for pointing out the incorrect sentence, and we will combine with the modification in comment 2.
# Page: 3 Ln: 88-89: As new catalysts of atomic scale, researchers had shown great interest, such as Fe, Co, Mn and other atoms as metal center atoms of SACs.
# For the new catalysts of atomic scale, researchers showed great interest in designing and preparing SACs using Fe, Co, Mn and other non-noble metal atoms.
And we read and check the manuscript carefully and revise it to improve the overall English level, and the modifications are marked in green.
[Comment 2] I have seen many review articles published on the topic of TM-N-C type ORR catalysts applied in PEFC cathodes. Why should another review be published on this topic? This has to be clearly described in the introduction section with emphasis on the points that are distinctly covered in the review and not in other similar reviews published earlier.
[Response] Thanks for your professional and detailed suggestions. We added content in the introduction section to make it more logical and point out the main points and purpose of this paper.
In order to advance the understanding and development of new high-performance non-noble metal catalysts, the research progress of non-noble metals and N co-doped carbon catalysts is extensively reviewed in this paper. However, many reviews have recently also been published [1-3]. Despite them, we not only subdivide non-noble catalysts into single-atom catalysts and double-atom catalysts, but also further focused on the methods and performances of preparing catalysts with Fe, Co and Mn as non-noble metal atoms. And the challenges and prospects of non-noble metal catalysts used in the ORR of PEMFCs are discussed and prospected. Specifically, the purpose and main contribution of this paper include: (i) The synthesis progress of non-noble metal catalysts (especially single-atom catalysts and double-atom catalysts) in the past five years was comprehensively summarized. (ii) The important things and challenges in the design and synthesis of non-noble metals were presented. This review can provide better insight into current progress and future directions, and provide some reference value for the related studies on the design and synthesis of non-noble metal catalysts.
[1] Gu, W.L.; Hu, L.Y.; Li, J.; Wang, E.K. Recent Advancements in Transition Metal-Nitrogen-Carbon Catalysts for Oxygen Reduction Reaction. Electroanalysis 2018, 30(7), 1217-1228.
[2] Chi, B.; Zhang, X.R.; Liu, M.R.; Jiang, S.J.; Liao, S.J. Applications of M/N/C analogue catalysts in PEM fuel cells and metal-air/oxygen batteries: Status quo, challenges and perspectives. Prog. Nat. Sci.: Mater. Int. 2020, 30(6), 807-814.
[3] Cui, J.Y.; Chen, Q.J.; Li, X.J.; Zhang, S.J. Recent advances in non-precious metal electrocatalysts for oxygen reduction in acidic media and PEMFCs: an activity, stability and mechanism study. Green Chem. 2021, 23(18), 6898-6925.
[Comment 3] The statement in the abstract and introduction section refers only to the PEMFC catalyst. However, in Table 1, the authors present the activity of the catalyst measured in alkaline pH. How this is relevant to PEMFC? The same comment holds true for Table 2. If the review is not exclusively focused on PEMFC, the statements in the appropriate sections using the acronym PEMFC should be deleted or modified.
[Response] Thanks for your professional and insightful comment. When we wrote the manuscript, we wanted to introduce the activity of the catalysts more comprehensively, so we added the activity in alkaline solution. As you said, it contradicts the theme of our paper. We will modify Table 1 and Table 2 according to your comments.
Table 1. Summary of the ORR performance for other metal SACs.
Catalysts |
Specific name |
E1/2 / Acid electrolyte |
Stability |
References |
Cu-SACs |
CuSA/CuCT@NPC |
0.80 V vs. RHE / 0.1 M HClO4 |
10000 cycles / 6 mV negative shift |
[94] |
Cu-SAs/NSs |
0.74 V vs. RHE / 0.1 M HClO4 |
3000 cycles |
[95] |
|
Ni-SACs |
Ni-N3-Gra |
Comparable with Pt |
- |
[96] |
Zn-SACs |
Zn-N-C |
0.746 V vs. RHE / 0.1 M HClO4 |
1000 cycles / 19.88 mV negative shift |
[97] |
Zn-B/N-C |
0.753 V vs. RHE / 0.1 M HClO4 |
Current density / mA·cm-2 (87 % remained after 80000 s) |
[98] |
|
A-Zn@NSG |
0.805 V vs. RHE / 0.1 M HClO4 |
5000 cycles / 6.7 mV negative shift |
[99] |
Table 2. Summary of the ORR performance for other metal DACs.
Catalysts |
Specific name |
E1/2 / Acid electrolyte |
Stability |
References |
FeCu-DACs |
FeCu-N-C |
0.784 V vs. RHE / 0.1 M HClO4 |
10000 cycles / 15 mV negative shift |
[142] |
FeCu/N-CNTs |
0.811V vs. RHE / 0.1 M HClO4 |
5000 cycles |
[143] |
|
FeCuNC |
0.820 V vs. RHE / 0.5 M H2SO4 |
- |
[144] |
|
FeNi-DACs |
FeNi-N6-C |
0.780V vs. RHE / 0.1 M HClO4 |
5000 cycles / Almost unchanged |
[145] |
Fe/Ni-NX/OC |
0.840 V vs. RHE / 0.1 M HClO4 |
5000 cycles / Almost unchanged |
[146] |
|
FeZn-DACs |
Fe-Zn-SA/NC |
0.780 V vs. RHE / 0.1 M HClO4 |
5000 cycles/ No obvious decrease |
[147] |
Zn/Fe–N–C |
0.810 V vs. RHE / 0.5 M H2SO4 |
40000 s / 85.6% current retained |
[148] |
|
CoNi-DACs |
CoPNi-N/C |
0.730 V vs. RHE / 0.1 M HClO4 |
5000 cycles / 11 mV negative shift |
[149] |
And we also deleted the literatures in the manuscript that FeMn-DAC only showed catalytic activity in alkaline environment.
[Comment 4] Conclusion section lacks clarity.
[Response] Thanks for your professional and pertinent comment. In order to make the conclusion more clear and logical, we have added two parts. One part is the important things in the design and preparation of non-noble metal catalysts, and the second part is the comparison and summary between the performance of current non-noble metal catalysts, the performance of commercial Pt/C catalysts, and the target performance of DOE.
In order to improve the output power, dynamic response, life and other comprehensive performance of PEMFCs, and accelerate their commercial process, it is urgent and meaningful to explore efficient and durable non-noble metal ORR catalysts. This paper mainly reviewed the research on non-noble metal ORR catalysts for PEMFCs in the past five years from the perspective of preparation and performance, which mainly included two categories: single transition metal atom catalyst and double transition metal atom catalyst. Generally, there are two main methods to synthesize non-noble metal ORR catalysts: (1) mixing and direct pyrolysis; (2) sacrificial-template method based on MOF, and then pyrolysis to obtain the catalyst. The precursor type, precursor structure, heat treatment time, heat treatment temperature and post-treatment operation in the preparation method will have an important impact on the activity and stability of the non-noble metal ORR catalysts. Furthermore, the surface area, active site and exposure rate of non-noble metal ORR catalysts directly affect the catalytic activity and stability. Therefore, in the design and preparation process of non-noble metal catalysts, it is important to select promising precursors, strictly control the heat treatment and post-treatment conditions, and strive to improve the surface area, active site and exposure rate of catalysts. Although great progress has been made in the preparation and performance of non-noble metal ORR catalysts for PEMFCs, there are still many challenges.
Firstly, there are many methods to synthesize SACs and DACs, but how to reduce the cost, shorten the synthesis cycle and improve the practicability of preparation methods is still a challenge. Secondly, in the preparation methods of SACs and DACs, how to precisely control the synthesis conditions and obtain catalysts with high surface area, multiple active sites and exposure remains to be further studied. Then, the research on SACs mainly focused on Fe, Mn, Co and Cu atoms, and the DACs mostly consisted of the above atoms, and the influence of the introduction of other transition metal atoms on the performance of TM-N-C catalysts still needs to be further explored. After that, the selection and use of C carriers can be further optimized or replaced to synthesize highly active and durable catalysts. Finally, most catalysts are difficult to exceed the overall performance of Pt/C catalysts in acidic conditions, and improving the activity and durability of catalysts in acidic conditions is still a challenge.
Undoubtedly, advanced non-noble metal catalysts have exhibited excellent activity and durability, showing similar performance to commercial Pt/C catalysts for fuel cell applications, and some non-noble metal catalysts even outperform 2020 target performance of DOE. Notably, the durability test of non-noble metal catalysts is carried out under laboratory conditions, which is still very different from the actual complex and changeable application scenarios. The durability of non-noble metal catalysts is still a challenge for commercial application of fuel cells. In the future, the design and preparation for ORR catalysts of PEMFCs should follow the comprehensive objectives of high activity, high durability, low price and scalability, and further optimize the preparation method to guide the realization of large-scale production and application of efficient and durable catalysts at an early date.
Once again, we would like to express our most sincere thanks for your careful reading and valuable comments.

Reviewer 3 Report
This manuscript reviewed and summarized the recent research developments of non-noble metal catalysts of oxygen reduction reaction (ORR). Explanations about properties of several transition metal catalysts are well organized. This paper provides good insight regarding non-noble metal ORR catalysts, but needs revisions by addressing the following issues.
1. Please add explanations about general advantages of single-atoms with respect to activity and stability. I think that there are lack of explanation about advantages of single-atoms and dual atoms catalysts compared to metal particle catalysts.
2. Please add explanations about anion exchange membrane fuel cells (AEMFCs) and proton exchange membrane fuel cells (PEMFCs). I think that single cell data are also important for real application of single atom based catalysts
3. Authors summarized in two parts (single atom and dual atom catalysts). However, both single atom cobalt and cobalt particles are formed on the support materials in reference [58], Therefore I think that that is not relevant for the first part. Addition of deeper discussion of actual roles of single atom cobalt is required.
4. I think that stability is important factor for real commercial applications. I think that this paper only focus on ORR activity enhancement. Please add explanation how single atom and dual atom enhance durability for ORR.
Author Response
Dear Reviewer,
We would like to express our sincere appreciation for your careful reading and invaluable comments to improve this paper. In order to effectively improve the quality of the paper, we have made the following modifications to your comments. The amendments made are mentioned below with reference to appropriate paragraphs and sections of the revised manuscript.
[Comment 1] Please add explanations about general advantages of single-atoms with respect to activity and stability. I think that there are lack of explanation about advantages of single-atoms and dual atoms catalysts compared to metal particle catalysts.
[Response] Thanks for your professional and serious suggestion. We add the explanation about advantages of SACs and DACs catalysts compared to metal particle catalysts in Section 2.1 on page 3~4.
The reduction of the size of metal particles is conducive to improving the reactivity of supported metal catalysts [47]. With the development of nanotechnology, the size of metal particles could be reduced to nanoscale or sub-nanoscale [48], and some reports indicated that sub-nanoscale supported metal catalysts can exhibit better catalytic activity [49,50]. The active sites exposure rate and catalytic activity of TM-N-C catalysts can be effectively improved by further reducing the non-noble metal nanoparticles to atomic scale [51,52]. Compared with nanoscale transition metal particle catalysts, atomic scale transition metal catalysts have many advantages: (i) with unique electronic structure and definite active site, the atomic scale catalysts can exhibit excellent catalytic performance [53,54]; (ii) the atomic scale catalysts can facilitate the activation of reactants by lowering energy barrier for a high selectivity [55,56]; (iii) from the perspective of atomic scale, the structure-performance relationship of catalysts can be clearly established and understanded, and with the help of DFT theory and experiments, the position of active sites can be clearly identified, which can provide reference for the improved design of high-performance atomic level transition metal catalysts [57,58].
[53] Gawande, M.B.; Fornasiero, P.; Zboril, R. Carbon-Based Single-Atom Catalysts for Advanced Applications. ACS Catal. 2020, 10(3), 2231-2259.
[54] Sun, K.; Xu, W.W.; Lin, X.; Tian, S.B.; Lin, W.-F.; Zhou, D.J.; Sun, X.M. Electrochemical Oxygen Reduction to Hydrogen Peroxide via a Two-Electron Transfer Pathway on Carbon-Based Single-Atom Catalysts. Adv. Mater. Interfaces 2020, 8(8), 2001360.
[55] Darby, M.T.; Stamatakis, M.; Michaelides, A.; Sykes, E.C.H. Lonely Atoms with Special Gifts: Breaking Linear Scaling Relationships in Heterogeneous Catalysis with Single-Atom Alloys. J. Phys. Chem. Lett. 2018, 9(18), 5636-5646. 7.301
[56] Xu, Z.L.; Ao, Z.M.; Yang, M.; Wang, S.B. Recent progress in single-atom alloys: Synthesis, properties, and applications in environmental catalysis. J. Hazard. Mater. 2021, 424, 127427. 12.984
[57] Zhang, T.J.; Walsh, A.G.; Yu, J.H.; Zhang, P. Single-atom alloy catalysts: structural analysis, electronic properties and catalytic activities. Chem. Soc. Rev. 2021, 50(1), 569-588.
[58] Ma, Y.L.; Jin, F.M.; Hu, Y.H. Bifunctional electrocatalysts for oxygen reduction and oxygen evolution: a theoretical study on 2D metallic WO2-supported single atom (Fe, Co, or Ni) catalysts. Phys. Chem. Chem. Phys. 2021, 23(24), 13687-13695.
And we add the explanations about general advantages of SACs with respect to activity and stability in Section 2.1 on page 4.
Single-atom catalysts (SACs) can maximize the utilization rate of transition metal atoms, theoretically reaching 100% of the atom utilization rate [59]. Moreover, the spatial structure of SACs is very uniform, with an unsaturated coordination environment and clear single atom sites, which can completely expose the active sites attached to the support surface [60]. At the same time, the unique electronic structure of transition metal active center atoms effectively improves catalytic activity and selectivity, as well as improve the stability of the catalysts [61,62]. These advantages provide the premise for the wide research and application of SACs. For the new catalysts of atomic scale, researchers showed great interest in designing and preparing SACs using Fe, Co, Mn and other non-noble metal atoms. For the new catalysts of atomic scale, researchers showed great interest in designing and preparing SACs using Fe, Co, Mn and other non-noble metal atoms.
[59] Shi, Z.S.; Yang, W.Q.; Gu, Y.T.; Liao, T.; Sun, Z.Q. Metal-Nitrogen-Doped Carbon Materials as Highly Efficient Catalysts: Progress and Rational Design. Adv. Sci. 2020, 7(15), 2001069.
[60] Tajik, S.; Dourandish, Z.; Nejad, F.G.; Beitollahi, H.; Afshar, A.A.; Jahani, P.M.; Di Bartolomeo, A. Review-Single-Atom Catalysts as Promising Candidates for Single-Atom Catalysts as Promising Candidates for Electrochemical Applications. J. Electrochem. Soc. 2022, 169(4), 046504.
[61] Li, J.; Yue, M.F.; Wei, Y.M.; Li, J.F. Synthetic strategies of single-atoms catalysts and applications in electrocatalysis. Electrochim. Acta 2022, 409, 139835.
[62] Chen, Y.J.; Ji, S.F.; Chen, C.; Peng, Q.; Wang, D.S.; Li, Y.D. Single-Atom Catalysts: Synthetic Strategies and Electrochemical Applications. Joule 2018, 2(7), 1242-1264.
[Comment 2] Please add explanations about anion exchange membrane fuel cells (AEMFCs) and proton exchange membrane fuel cells (PEMFCs). I think that single cell data are also important for real application of single atom based catalysts.
[Response] Thanks for your professional and pertinent suggestions. Because the title and abstract of our manuscript are related to proton exchange membrane fuel cells (PEMFCs), it is not appropriate for us to add relevant explanations to anion exchange membrane fuel cells in this review. At the same time, we suspect that the performance of alkaline electrolyte in Table 1 and Table 2 may mislead you. We are very sorry for this, and we have modified Table 1 and Table 2 to be closer to the application of PEMFCs in the title and abstract.
Table 1. Summary of the ORR performance for other metal SACs.
Catalysts |
Specific name |
E1/2 / Acid electrolyte |
Stability |
References |
Cu-SACs |
CuSA/CuCT@NPC |
0.80 V vs. RHE / 0.1 M HClO4 |
10000 cycles / 6 mV negative shift |
[100] |
Cu-SAs/NSs |
0.74 V vs. RHE / 0.1 M HClO4 |
3000 cycles |
[101] |
|
Ni-SACs |
Ni-N3-Gra |
Comparable with Pt |
- |
[102] |
Zn-SACs |
Zn-N-C |
0.746 V vs. RHE / 0.1 M HClO4 |
1000 cycles / 19.88 mV negative shift |
[103] |
Zn-B/N-C |
0.753 V vs. RHE / 0.1 M HClO4 |
Current density / mA·cm-2 (87 % remained after 80000 s) |
[104] |
|
A-Zn@NSG |
0.805 V vs. RHE / 0.1 M HClO4 |
5000 cycles / 6.7 mV negative shift |
[105] |
Table 2. Summary of the ORR performance for other metal DACs.
Catalysts |
Specific name |
E1/2 / Acid electrolyte |
Stability |
References |
FeCu-DACs |
FeCu-N-C |
0.784 V vs. RHE / 0.1 M HClO4 |
10000 cycles / 15 mV negative shift |
[142] |
FeCu/N-CNTs |
0.811V vs. RHE / 0.1 M HClO4 |
5000 cycles |
[143] |
|
FeCuNC |
0.820 V vs. RHE / 0.5 M H2SO4 |
- |
[144] |
|
FeNi-DACs |
FeNi-N6-C |
0.780V vs. RHE / 0.1 M HClO4 |
5000 cycles / Almost unchanged |
[145] |
Fe/Ni-NX/OC |
0.840 V vs. RHE / 0.1 M HClO4 |
5000 cycles / Almost unchanged |
[146] |
|
FeZn-DACs |
Fe-Zn-SA/NC |
0.780 V vs. RHE / 0.1 M HClO4 |
5000 cycles/ No obvious decrease |
[147] |
Zn/Fe–N–C |
0.810 V vs. RHE / 0.5 M H2SO4 |
40000 s / 85.6% current retained |
[148] |
|
CoNi-DACs |
CoPNi-N/C |
0.730 V vs. RHE / 0.1 M HClO4 |
5000 cycles / 11 mV negative shift |
[149] |
And we also deleted the literatures in the manuscript that FeMn-DAC only showed catalytic activity in alkaline environment.
[Comment 3] Authors summarized in two parts (single atom and dual atom catalysts). However, both single atom cobalt and cobalt particles are formed on the support materials in reference [58], Therefore I think that that is not relevant for the first part. Addition of deeper discussion of actual roles of single atom cobalt is required.
[Response] Thanks for your professional and insightful comment, and we agree with you very much. According to your suggestion, we searched the relevant literature again and supplemented the introduction of single atom cobalt to make our review more comprehensive.
Wan et al. reported that the ORR catalysts synthesized based on ZIF had the problems of large particle size and low mesoporous ratio, leading to poor electron conductivity and affecting the catalytic performance of ORR [82]. Therefore, increasing mesoporous rate and conductivity is an effective strategy to improve ORR catalyst [83]. Wang et al. synthesized a CoNC@KJ600 catalyst with high pore structure and high electronic conductivity based on ZIF, and used the same procedure to synthesize CoNC catalyst to compare and verify the performance of CoNC@KJ600 [84]. The synthesis procedure was shown in Figure. 5(a). During the synthesis process, the porous structure of KJ600 carbon black was retained, and the Co element was highly dispersed in CoNC@KJ600 catalyst. However, there were lots of Co nanoparticles in CoNC catalyst. The presence of Co nanoparticles would block the mass transfer gap and reduce the activity of ORR catalyst [85]. As shown in Figure 5(b) and Figure 5(c), the catalytic current density of CoNC@KJ600 catalyst was slightly higher than that of CoNC catalyst (1.58 vs. 1.28 A g-1 @ 0.8V), and CoNC@KJ600 catalyst was more durable than CoNC catalyst after 20 h test. Considering the high pore structure and high electronic conductivity of CoNC@KJ600 catalyst, CoNC@KJ600 catalyst could be applied to PEMFC. The peak power density of PEMFC with CoNC@KJ600 catalyst as cathode was 920 W/cm2, which was higher than that reported by XX et al. [86] and YY et al. [87] for PEMFC with Co-N-C catalyst as cathode.
Figure 5. Synthesis method and performance of CoNC@KJ600 and CoNC catalysts: (a) Synthesis diagram; (b) ORR activity; (c) ORR durability; (d) PEMFC performance. Adapted from Ref. [84]. Copyright 2020 Elsevier.
[82] Wan, X.; Liu, X.F.; Li, Y.C.; Yu, R.H.; Zheng, L.R.; Yan, W.S.; Wang, H.; Xu, M.; Shui, J.L. Fe-N-C electrocatalyst with dense active sites and efficient mass transport for high-performance proton exchange membrane fuel cells. Nat. Catal. 2019, 2(3), 259-268.
[83] Xia, W.; Zou, R.Q.; An, L.; Xia, D.G.; Guo, S.J. A metal-organic framework route to in situ encapsulation of Co@Co3O4@C core@ bishell nanoparticles into a highly ordered porous carbon matrix for oxygen reduction. Energy Environ. Sci. 2015, 8(2), 568-576.
[84] Wang, R.X.; Zhang, P.Y.; Wang, Y.C.; Wang, Y.S.; Zaghib, K.; Zhou, Z.Y. ZIF-derived Co-N-C ORR catalyst with high performance in proton exchange membrane fuel cells. Prog. Nat. Sci.: Mater. Int. 2021, 30(6), 855-860.
[85] Kramm, U.I.; Herrmann-Geppert, I.; Behrends, J.; Lips, K.; Fiechter, S.; Bogdanoff, P. On an Easy Way To Prepare Metal Nitrogen Doped Carbon with Exclusive Presence of MeN4-type Sites Active for the ORR. J. Am. Chem. Soc. 2016, 138(2), 635-640.
[86] Cheng, Q.Q.; Han, S.B.; Mao, K.; Chen, C.; Yang, L.J.; Zou, Z.Q.; Gu, M.; Hu, Z.; Yang, H. Co nanoparticle embedded in atomically-dispersed Co-N-C nanofibers for oxygen reduction with high activity and remarkable durability. Nano Energy 2018, 52, 485-493.
[87] Im, K.; Jang, J.H.; Heo, J.; Kim, D.; Lee, K.S.; Lim, H.-K.; Kim, J.; Yoo, S.J. Design of Co-NC as efficient electrocatalyst: The unique structure and active site for remarkable durability of proton exchange membrane fuel cells. Appl. Catal., B 2022, 308, 121220.
[Comment 4] I think that stability is important factor for real commercial applications. I think that this paper only focus on ORR activity enhancement. Please add explanation how single atom and dual atom enhance durability for ORR.
[Response] Thanks for your professional opinion. In section 3, We point out that the durability of non-noble metal catalysts is still a challenge for commercial application of fuel cells. As you suggested, we did not explain how single atom and dual atom enhance durability for ORR. We thank you very much for your comment and added relevant explanation at the end of section 2 on page 15.
The ORR durability of non-noble metal catalysts is of great value for real commercial applications. However, above studies only tested ORR durability in the laboratory environment and did not consider measures to improve ORR durability of non-noble metal catalysts. Atomic scale metal elements have high surface energy, which will make single metal atoms tend to aggregate and destroy the stability of SACs [150,151]. For DACs, the introduction of metal atoms in different d-band can effectively adjust the electronic structure and improve the ORR durability of the catalysts [152,153]. Therefore, studies on the durability of non-noble metal catalysts mainly focus on the improvement of the stability of SACs [154-156]. Wang et al. concluded that defect-anchoring strategy and confinement strategy were the two most common stability strategies, which could enhance the interaction between metal atoms and support [157,158]. For example, Abdul Majid et al. reported that single Cu atoms anchoring and capping defect sites on the Zr oxide clusters of UiO-66 could improve the stability of Cu/ UiO-66 catalysts [159]. The effective interaction between single metal atoms and the support can not only avoid the clusters between atoms, but also regulate the electronic structure of the catalysts [160]. Therefore, the surface and microstructure of the support are the key factors to improve the stability of the SACs, which is also relatively easy to realize and control.
[150] Corma, A.; Concepcion, P.; Boronat, M.; Sabater, M.J.; Navas, J.; Yacaman, M.J.; Larios, E.; Posadas, A.; Lopez-Quintela, M.A.; Buceta, D.; Mendoza, E.; Guilera, G.; Mayoral, A. Exceptional oxidation activity with size-controlled supported gold clusters of low atomicity. Nat. Chem. 2013, 5(9), 775-781.
[151] Cheng, N.C.; Zhang, L.; Doyle-Davis, K.; Sun, X.L. Single-Atom Catalysts: From Design to Application. Electrochem. Energy Rev. 2019, 2(4), 539-573.
[152] Yan, Y.; Cheng, H.Y.; Qu, Z.H.; Yu, R.; Liu, F.; Ma, Q.W.; Zhao, S.; Hu, H.; Cheng, Y.; Yang, C.Y.; Li, Z.F.; Wang, X.; Hao, S.Y.; Chen, Y.Y.; Liu, M.K. Recent progress on the synthesis and oxygen reduction applications of Fe-based single-atom and double-atom catalysts. J. Mater. Chem. A 2021, 9(35), 19489-19507.
[153] Ma, M.; Kumar, A.; Wang, D.N.; Wang, Y.Y.; Jia, Y.; Zhang, Y.; Zhang, G.X.; Yan, Z.F.; Sun, X.M. Boosting the bifunctional oxygen electrocatalytic performance of atomically dispersed Fe site via atomic Ni neighboring. Appl. Catal. B-Environ 2020, 274, 119091.
[154] Luo, X.; Wei, X.Q.; Wang, H.J.; Gu, W.L.; Kaneko, T.; Yoshida, Y.; Zhao, X.; Zhu, C.Z. Secondary-Atom-Doping Enables Robust Fe-N-C Single-Atom Catalysts with Enhanced Oxygen Reduction Reaction. Nano-Micro Lett. 2020, 12(1), 163.
[155] Yang, W.J.; Zhao, M.L.; Ding, X.L.; Ma, K.; Wu, C.C.; Gates, I.D.; Gao, Z.Y. The effect of coordination environment on the kinetic and thermodynamic stability of single-atom iron catalysts. Phys. Chem. Chem. Phys. 2020, 22(7), 3983-3989.
[156] Wan, X.; Shui, J.L. Exploring Durable Single-Atom Catalysts for Proton Exchange Membrane Fuel Cells. ACS Energy Lett. 2022, 7(5), 1696-1705.
[157] Wang, Y.X.; Cui, X.Z.; Zhang, J.Q.; Qiao, J.L.; Huang, H.T.; Shi, J.L.; Wang, G.X. Advances of atomically dispersed catalysts from single-atom to clusters in energy storage and conversion applications. Prog. Mater. Sci. 2022, 128, 100964.
[158] Jiao, L.; Zhang, R.; Wan, G.; Yang, W.J.; Wan, X.; Zhou, H.; Shui, J.L.; Yu, S.-H.; Jiang, H.-L. Nanocasting SiO2 into metal–organic frameworks imparts dual protection to high-loading Fe single-atom electrocatalysts. Nat. Commun. 2020, 11(1), 2831.
[159] Abdel-Mageed, A.M.; Rungtaweevoranit, B.; Parlinska-Wojtan, M.; Pei, X.K.; Yaghi, O.M.; Behm, R.J. Highly Active and Stable Single-Atom Cu Catalysts Supported by a Metal-Organic Framework. J. Am. Chem. Soc. 2019, 141(13), 5201-5210.
[160] Ji, S.F.; Chen, Y.J.; Wang, X.L.; Zhang, Z.D.; Wang, D.S.; Li, Y.D. Chemical Synthesis of Single Atomic Site Catalysts. Chem. Rev. 2020, 120(21), 11900-11955.
Once again, we would like to express our most sincere thanks for your careful reading and valuable comments.

Reviewer 4 Report
This work reviews the recent progress in the synthesis and performance of single and double atoms catalyst using Nitrogen heterogenous catalyst for ORR process. The work is publishable after the minor suggestion below:
1. The title of the paper appears to describe a much broader context than the review's content. As a result, I would advise the author to be more specific in the title.
2. The authors should hire an outside English editing service. I get the impression that the style is "translational," with a lot of long sentences.
Author Response
Dear Reviewer,
We would like to express our sincere appreciation for your careful reading and invaluable comments to improve this paper. In order to effectively improve the quality of the paper, we have made the following modifications to your comments. The amendments made are mentioned below with reference to appropriate paragraphs and sections of the revised manuscript.
[Comment 1] The title of the paper appears to describe a much broader context than the review's content. As a result, I would advise the author to be more specific in the title.
[Response] Thanks for your professional and pertinent suggestions. When we wrote the manuscript, we wanted to introduce the activity of the catalysts more comprehensively, so we added the activity in alkaline solution. However, this led to the fact that part of the contents of the manuscript were not completely consistent with the title and abstract. We suspect that the performance of alkaline electrolyte in Table 1 and Table 2 may mislead you. We are very sorry for this, and we have modified Table 1 and Table 2 to be closer to the application of PEMFCs in the title and abstract.
Table 1. Summary of the ORR performance for other metal SACs.
Catalysts |
Specific name |
E1/2 / Acid electrolyte |
Stability |
References |
Cu-SACs |
CuSA/CuCT@NPC |
0.80 V vs. RHE / 0.1 M HClO4 |
10000 cycles / 6 mV negative shift |
[100] |
Cu-SAs/NSs |
0.74 V vs. RHE / 0.1 M HClO4 |
3000 cycles |
[101] |
|
Ni-SACs |
Ni-N3-Gra |
Comparable with Pt |
- |
[102] |
Zn-SACs |
Zn-N-C |
0.746 V vs. RHE / 0.1 M HClO4 |
1000 cycles / 19.88 mV negative shift |
[103] |
Zn-B/N-C |
0.753 V vs. RHE / 0.1 M HClO4 |
Current density / mA·cm-2 (87 % remained after 80000 s) |
[104] |
|
A-Zn@NSG |
0.805 V vs. RHE / 0.1 M HClO4 |
5000 cycles / 6.7 mV negative shift |
[105] |
Table 2. Summary of the ORR performance for other metal DACs.
Catalysts |
Specific name |
E1/2 / Acid electrolyte |
Stability |
References |
FeCu-DACs |
FeCu-N-C |
0.784 V vs. RHE / 0.1 M HClO4 |
10000 cycles / 15 mV negative shift |
[142] |
FeCu/N-CNTs |
0.811V vs. RHE / 0.1 M HClO4 |
5000 cycles |
[143] |
|
FeCuNC |
0.820 V vs. RHE / 0.5 M H2SO4 |
- |
[144] |
|
FeNi-DACs |
FeNi-N6-C |
0.780V vs. RHE / 0.1 M HClO4 |
5000 cycles / Almost unchanged |
[145] |
Fe/Ni-NX/OC |
0.840 V vs. RHE / 0.1 M HClO4 |
5000 cycles / Almost unchanged |
[146] |
|
FeZn-DACs |
Fe-Zn-SA/NC |
0.780 V vs. RHE / 0.1 M HClO4 |
5000 cycles/ No obvious decrease |
[147] |
Zn/Fe–N–C |
0.810 V vs. RHE / 0.5 M H2SO4 |
40000 s / 85.6% current retained |
[148] |
|
CoNi-DACs |
CoPNi-N/C |
0.730 V vs. RHE / 0.1 M HClO4 |
5000 cycles / 11 mV negative shift |
[149] |
And we also deleted the literatures in the manuscript that FeMn-DAC only showed catalytic activity in alkaline environment.
[Comment 2] The authors should hire an outside English editing service. I get the impression that the style is "translational," with a lot of long sentences.
[Response] Thanks for your professional and serious comment. And we will read and check the manuscript carefully and revise it to improve the overall English level, and the modifications are marked in green.
Once again, we would like to express our most sincere thanks for your careful reading and valuable comments.

Round 2
Reviewer 1 Report
Thanks you for the response on my questions. I think the revised version is now enough to be published in Nanomaterials.
Reviewer 2 Report
The authors have addressed all the comments provided by me. The revised version of the review article can be considered for publication.